# Designing Categories for a Mixed-Method Research on Competence Development and Professional Identity through Collegial Advice in Nursing Education in Germany

**DOI:** 10.3390/healthcare10122517

**Published:** 2022-12-12

**Authors:** Stefan Wellensiek, Jan P. Ehlers, Michaela Zupanic

**Affiliations:** 1Bildung & Beratung Bethel, 33604 Bielefeld, Germany; 2Didactics and Educational Research in Health Science, Faculty of Health, Witten/Herdecke University, 58455 Witten, Germany; 3Interprofessional and Collaborative Didactics in Medical and Health Degree Programs, Faculty of Health, Witten/Herdecke University, 58455 Witten, Germany

**Keywords:** competence development, professional identity, collegial advice, nursing education qualitative content analysis, category system

## Abstract

Background: The aim of nursing education in Germany is the development of different competences, including professional identity. To promote this, the use of collegial consultation in the form of collegial advice is recommended. How collegial advice affects the development of competences and professional identity, and which didactic and organizational framework conditions are favorable for this have not yet been conclusively clarified. Objectives: The aim of the study is to determine how collegial consultation affects the development of competence and professional identity of student nurses. Enabling and hindering factors for the success of collegial advice will be identified. Design/ Participants: A mixed-methods study with 25 student nurses who completed training in collegial advice and then regularly engaged in collegial advice for one year. Methods: A content analysis from four focus group interviews using a category system developed for this purpose. Results: This article reports the development of the category system necessary for the content analysis with examples. The resulting categories are presented. Conclusions: The category system has high objectivity, reliability, and validity. It contains links to competence and identity research in the care sector. A suitable instrument has been developed for further evaluation of the focus group interviews.

## 1. Introduction and Background

To foster personal fulfilment, health, employability and social inclusion, lifelong learning and the development of competences has become a central aim in education [1]. In the education of nurses, lifelong learning for the continuous development of different skills, e.g., reflection or communication is a major expectation [2]. Patients expect professional nurses to be competent in their practice and in the care they provide [3].

In Germany, nursing staff is mainly qualified through vocational training. This is divided into theoretical training at a nursing schools and practical training in various fields of nursing, e.g., hospitals or old people’s homes. This lasts three years, ends with a state examination and is regulated by the Act on the Nursing Professions. The aim of the vocational training is the development of professional competences. This means to gain the ability and willingness to act professionally in complex nursing situations [4]. In this context, competences are understood as interaction of skills, abilities, experience, and knowledge [5]. Another point in the development of competences is the capacity for self-reflection [6]. This means to reflect on and explain one’s own actions on legislation, regulations, scientific findings and professional ethical values and attitudes. Inherent in this is to apply basic conflict-resolution principles, to ask for, to accept and draw on the advice of colleagues [7].

Furthermore, the development of professional identity in the context of training is of importance [6]. According to Rauner, Evans and McLennon [8] professional identity is seen as a part of professional competence. Without the development of professional identity, no competence development would be conceivable [8]. Professional identity is composed of different elements: interest in the content of professional tasks, identification with the occupation, an experience-based occupational profile, reflected work experience and the classification of a professional role in the company’s business process [8]. The development of a professional identity is influenced by others, especially other nurses who are important as role-models and peers in this process [9]. 

In order to be able to develop self-reflective competence and professional identity, it therefore seems to make sense to give trainees the opportunity to reflect on their professional experiences on an ongoing basis in a peer group. One tool for this is collegial consultation. Generally, collegial consultation refers to reciprocal reflection among colleagues with the aim of generating new impetus for their daily work [10]. 

Various formats of collegial consultation in a nursing setting are possible. A very pragmatic form is curbside consulting. Curbside consulting is a process used by nurses to informally seek advice from one another without requesting formal consultation. It offers bidirectional or conversational direct answers to specific questions at the Point-of-care [11]. Another well-known form is clinical supervision. It is described as “a formal process of professional support and learning which enables individual practitioners to develop knowledge and competence, assume responsibility for their own practice and enhance consumer protection and safety of care in complex clinical situations.” [12], (p. 472). It has widespread acceptance within the nursing profession. There are various models of clinical supervision to support the clinical learning environment, including group or 1:1 models [13]. All these models involve observation through a clinical supervisor. Another possibility of collegial consultation is the use of Balint groups. In Balint groups, six to eight colleagues reflect on their experiences and emotional aspects of their work. Balint groups are led by a professional, e. g. a psychotherapist [14]. Complementary to these types of collegial consultation, numerous models for reflection have been published, e.g., Schön (reflective-practice concept) [15], Johns (model for structured reflection) [16] or Korthagen (ALACT-model) [17]. 

In contrast to all these models, collegial advice, also known as intervision [18], is a consultation process that is carried out in a group of colleagues, in which participants discuss professional practice issues by following a specified process with distributed and reversible roles (e.g., case narrator, coach, moderator) [19]. The main criteria defining collegial advice are peer-led groups, no need for external facilitators, specified roles, reciprocity of all roles and focus on professional practice situations. Collegial advice is relatively well established in continental Europe [20]. Various positive effects in different fields of work are described. Norcross, Geller and Kurzawa [21] summarize that collegial advice leads to a better perception of the role and identity. Roy, Genest Dufault and Châteauvert [22] report on a heterogenous collegial advice group with expert and novice teachers. They could solve problems and help each other to focus their roles. Tietze [19] demonstrates that the group members’ experiences and attitudes can lead to problem solving by experiencing parallels between their own experiences and those of the peer group. This allows them to rethink own attitudes and derive or reflect on solution options for similar problems or conflict situations. Roddewig [23] focuses on collegial advice in nursing training. She demonstrates positive effects: transfer of specialist knowledge, development of perceptual skills, learning empathy, active listening, feedback, and congruent behaviour, with professionalization through regular reflection on one’s own role and improvement of competence by the discussion of solution options. Collegial advice in a peer group offers various advantages, can be introduced in educational settings, and then be transferred into work life. 

There are thus some references that collegial advice has the potential to foster the develop of competences and professional identity. Whether this is also the case in nursing training has not yet been conclusively clarified. Little is known about conducive and inhibiting organizational and didactic framework conditions in relation to collegial advice in nursing education.

The research on which this report is based therefore investigates the question of whether collegial advice can be implemented in vocational training of student nurses and made usable for the purpose of reflecting on professional experiences. In order to be able to make concrete recommendations for, e.g., the development of curricula or concepts, another aspect of the study is the question of which didactic and organizational framework conditions have a beneficial or inhibiting effect on the success of collegial consulting. Therefore, this article reports on the development of a category system that represents the influence of collegial advice on the development of professional competences and identity and various framework conditions. As a result, a category system is presented, which is assessed according to the main quality criteria of objectivity, reliability, and validity.

## 2. Methods

### 2.1. Participants

The study took place in an educational centre for health care professions with a group of student nurses. The group consisted of n = 25 persons (23 w/2 m) aged 19–25 years with school-leaving qualifications after 10, 12 or 13 years. They had already completed one year of training with several theoretical and practical teaching-learning experiences together. Identified drop-out risks such as permanent illness or pregnancy did not occur during the study period. During a preparatory training, the student nurses and the researcher got to know each other well, so they were familiar with each others’ biographies, qualifications, and motivations. In this context, the researcher acted as a trainer for collegial advice. Prior to the start of the research, there was no relationship between the researcher and the student nurses. Before the intervention began, the research group was familiarized with the objectives and instruments of the research project.

### 2.2. Intervention

In the period from October to December 2017, a structured training of 18 lessons on the topic of collegial advice took place. The participants received a brochure containing, for example, a description of five consulting methods and agreements on cooperation. Subsequently, four random groups (G1-G4) were formed, each with six to seven student nurses. The study period extended from February to November 2018. Each group met seven times (t1 to t7). To gain methodological confidence, the first collegial consultation of all groups (t1) was conducted together with the researcher as moderator (=external moderation) in each case. Groups G1 and G2 conducted all subsequent consultations (t2-t7) also with external moderation. Groups G3 and G4 worked without accompaniment by the researcher or other experts and moderated the meetings independently (=internal moderation). Figure 1 shows a schematic of the procedure of the intervention.

### 2.3. Data Collection

The group meetings took place at intervals of eight to ten weeks. The duration, topics, methods, and key questions of the consultation were recorded in a standardized protocol by the participants in each meeting. The topics are mainly related to the research question, if collegial advice is usable for the purpose of reflecting on professional experiences. Didactic and organizational framework conditions appear as a secondary aspect. After the group sessions, a focus group interview was conducted with each group in January 2019. A linear interview guide was developed to ensure structural comparability of the focus group interviews (Appendix A). The data recorded in the group meeting, e.g., the duration of the meetings and the use of the methods, reappear in the interview guide. The focus group interview incorporates all research questions. The aim was to capture the students’ perspectives on the past consultations and to map individual impressions. The themes of the interviews are partly derived from an analysis of the existing literature. For example, the type of moderation is seen as a didactic condition [23]. Other topics follow the further research interest, e.g., whether nursing students can reflect real professional experiences in the context of collegial counselling and what recommendations they can make for curricular integration. To open the interview, general impressions of past consultations were elicited. Then the didactic and organizational circumstances were reviewed. Specifically, the number of the professional situations described and worked on, the distribution of roles, the presence of participants, the use of methods, and the duration of meetings were discussed. The participants then discussed the question of whether collegial consultation is suitable for reflecting on professional experiences. The focus was on the key questions that each group consulted on. Then, participants were asked to analyze possible influences on the deliberations, e.g., well-being, the presence or absence of a facilitator, or the usefulness of the brochure, and to make recommendations for implementing collegial advice. Finally, participants evaluated whether and why it makes (no) sense to establish collegial advice in education. These questions target perceptions of the opportunity for self-reflection and identity development, as well as the peer group experience. The interviews were audio-recorded. Supplementary field notes were not made. As a result, four focus group interviews with a length between 75 and 93 min were conducted. No interview was terminated, although saturation of the content occurred in some cases.

### 2.4. Data Processing

In the first step, transcripts of the focus group interviews were created. These were written directly in MAXQDA 2018/2. To ensure that transcripts are systemic and consistent, they should be based on transcription guidelines [24]. The guidelines for this study include the requirements of the software and are easy-to-learn, but sufficient rules, e.g., “Each speech contribution is transcribed as a separate paragraph” or “Speech is transcribed verbatim, i.e., not phonetically or in summary form” [25], (p. 42). During transcription, 253 ad hoc memos were created for the individual paragraphs. These contained, e.g., notes for the formation of categories, questions and hypotheses that could be useful for the further course of category formation.

### 2.5. Data Analysis

Oriented on Kuckartz and Rädiker [25] and Döring and Bortz [26] a mixed form between deductive and inductive category formation was used. The initial creation of the categories drew upon the research questions and first impressions from the transcripts of the interviews. Figure 2 provides an overview of the process of creating the category system in the preparation phase and then the four research steps. 

Step 1: Deductive category formation

Based on the research questions, first impressions from the interviews and the transcripts, a priori categories [25] were formed. Four superordinate categories reflect the content of the research question. They are intended for structuring purposes only and will not contain any codes. Then, 12 main categories are created. Five of the main categories are further subdivided by a total of 16 subcategories. This preliminary category system (version 1) was created in MAXQDA. All categories contain the five elements: name, description and application of the category, example of application and differentiation from other categories. Thus, they are precisely formulated and can be used and discussed by different coders [25]. 

Step 2: First testing of the category system

In order to test the suitability of the category system, the procedure of O’Connor and Joffe [27] is used in principle. They describe guidance on the practical elements of performing an intercoder reliability assessment. 

To test the version 1 of the category system, the transcript of the interview of group G1 was used as material for coding. The smallest coding unit was defined as one sentence, the largest coding unit as several coherent sentences of an interviewee. A text passage that appeared relevant to the research questions and corresponded to the coding units is referred to as a segment. The double coding of a segment was accepted because a segment can contain several topics [28]. Only in this way is an accurate coding of statements of the interviewees possible.

The first coder (SW) identified 154 segments and then coded them. There was no duplicate coding. Then the work was paused for eight weeks in order to be able to reassess and revise the result with a certain distance and to reflect on one’s own attitude [25]. After the break, the previous result was put to the test. As a result, the categories were revised in terms of language and content and re-sorted. After all coding was deleted, the transcript was recoded, retaining the existing segments. Twenty-three new segments were identified. A total of 177 codes resulted. All but three of the main categories and subcategories were used. No new categories emerged during the coding process. 

In parallel, another person (JW) independently created a second category system. She was familiar with the method of collegial advice, knew the objective of the research and was present in two training sessions. She is experienced in qualitative research and comes from a complementary discipline. Therefore she matches the criteria of a second coder [29]. She was given various materials (e.g., research questions, interview guide, transcript G1). The category system she formed was written down. 

Step 3: Triangulation to ensure reliability

In the sense of triangulation to ensure reliability, the agreement of the two category systems between SW and JW was discussed. The aim was a qualitative examination of the chosen categories. For this purpose, different terms were first clarified. In the discussion it could be revealed that the choice of words was partly different, but the meant objects were the same. The contents of the systems were presented in a synopsis. Weaknesses were uncovered and corrected. Problems of individual categories were named and corrected. Overall, there was a high level of agreement between the systems in terms of language and content. The discussion resulted in a consensus category system (version 3) with more precise category definitions. Finally, both coders performed an independent re-coding in the category system version 3 with the existing segments.

Step 4: testing of the intercoder reliability

Intercoder reliability is described as „a numerical measure of the agreement between different coders regarding how the same data should be coded“ [27], (p. 2). In the testing, attention was focused on the question “[…] to what extent two people identify the same topics, aspects, and phenomena in the data and assign these to the same categories“ [25], (p. 267). The two coded interviews created in step 3 were used as material. First, it was considered whether the coding units and segment boundaries were correctly determined. It was found that 176 of 177 segments were used by both coders. This represents a 99.43% match and is a strong indication that the segments and their boundaries are appropriate. Four new segments were identified and accepted as useful additions and clarifications. There were no effects on the category system.

Subsequently, the extent to which both coders assigned a segment to the same category was checked. A total match of 74.19% (276 matches, 96 non-matches) was achieved. In total, there were six categories in which all codes matched (100%). Six categories had very high levels of agreement (80–96.55%). Two categories had seemingly low agreement scores (consulting method “actstorming” 66.67%, consulting method “upside-down” 66.67%). However, for these categories, the total number of segments assigned (3 and 6) is so small that even one or two non-matches result in values far below 80%. For these 14 categories in total, the coding instructions seem to be clear, and segments can be assigned with confidence. Four categories had 0% agreement. Ten other categories had unsatisfactory values ranging from 28.57% to 78.57%. 

The following step to evaluate the intercoder agreement is the calculation of a randomly corrected coefficient (kappa) on segment level. The kappa Κ_n_ was calculated according to Brennan and Prediger [30]. A total of 372 segments (276 matches, 96 mismatches) are included in the calculation. The value before further revision is 0.73. This value is considered as “fair” [31], (p. 372). To improve the result, the differences were followed up in an analytical peer discussion of the two coders SW and JW. This involved contrasting the non-matches and looking at each segment under the question of why the coders coded it the way they did and whether the category description was misleading. Three reasons for discrepancies in coding were identified. Some segments were coded in a main category instead of a subcategory. This indicated that the definitions of these categories needed to be clarified. Sometimes a segment contains different topics and could be coded single or multiple times. SW coded all segments only once. JW coded a total of seven segments twice. Another finding from the discussion was that the content of the segments was interpreted differently. This is understood as a hint to clarify terms and to consider the context of the segments in the further course of coding. Through this approach, each disagreement was investigated, and the cause identified to improve the category system. Categories were clarified in language and partially rewritten to provide clearer coding instructions to coders. 

After the agreed changes, the kappa K_n_, according to Brennan and Prediger [30], was calculated again. A total of 378 segments (316 matches, 62 mismatches) were now included in the calculation. The kappa increased to 0.83. According to Landis and Koch [31]. (p. 372) this value appears as “almost perfect”. Therefore, the category system is considered finalized from this point on.

## 3. Results

All comments and results arising during the analysis have been recorded, discussed, and incorporated into the optimization of the category system and the coding instructions. The finalized category system (version 4) is presented in Figure 3.

Table 1 gives an insight into the category system. The left column contains the names of the categories. These are divided into four main groups (organisational conditions, didactic conditions, recommendations for the implementation of collegial advice, reflection on professional experiences) with different subgroups. The middle column differentiates whether it is a superordinate category, main category, or subcategory. The right column presents anchor examples to each category. These were taken from the coding and represent a typical example. The anchor examples were included in the category descriptions in order to make the coding instructions more precise.

Table 2 represents a complete category description by the example of the subcategory 4.1.2. This category should help to find out if collegial advice can made usable for the purpose of reflecting on professional experiences. 

## 4. Discussion and Conclusions

Finally, the question of the main quality criteria of objectivity, reliability, and validity [32] can be considered.

High objectivity, i.e., the independence of the measurement results from the person collecting and evaluating these data, is achieved by clarified terms, the definition of coding units, segments, and coding rules as well as clearly defined categories with anchor examples. In this way, the coding result becomes highly independent of subjective impressions of the coders. Due to the elaborate and small-step documented process, the category system and the underlying descriptions appear reliable and valid, oriented to Villiger, Schweiger and Baldauf [29]. The measure of reliability, in this case intercoder reliability, is represented by the determination of the Kappa [30]. Very good results were obtained here. Other factors that speak for a high reliability of the instrument are the clear definitions and the delimitability of the individual categories. Thus, random or erroneous classifications become unlikely. 

When considering validity, the focus is on construct validity. The question arises to what extent the category system represents the constructs of professional identity and competence development. Categories consider elements of competence development, e.g., the capacity of self-reflection and self-organization (2.4). Other categories are focused on capturing identity, e.g., to make reflections on work experience (4.1.2 and 4.1.3), the classification of a professional role (4.1) or working in peer groups (1.3, 1.4, 2.1, 4.2). Further categories are directed to the organizational and didactic conditions and the recommendations for the implementation of collegial advice, and thus to the further research questions of the study. The validity of the instrument seems to be high here. A final assessment can only be made after the further transcripts are coded and related to the quantitative data. 

An application of the category system to research in other professions is limited. It only seems possible if the prevailing understanding of competence and identity and the structures there are comparable. In Germany this is the case for vocational training in other health professions, e.g., physiotherapy. It is not the case for, e.g., technological professions or university degrees. Working environment, learning conditions, professional identity and understanding of competences are different. This is where the instrument reaches its limits. Otherwise, the validity of the construct must be questioned.

The category system can additionally be considered in relation to other research. Since there is no research on collegial advice in nursing education in the international framework so far, only connections to the national literature can be drawn here. The subcategories contained in superordinate category 4 “Reflection on professional experiences” reflect different intentions of the Act on the Nursing Professions. The capacity to self-reflect is particularly important here. The superordinate categories 1–3 provide concrete guidance for the development of curricula as required by the Act on the Nursing Professions. Should collegial consultation prove to be a suitable measure for the development of competence and identity, more concrete measures for theoretical and practical teaching could be planned.

In her quantitatively oriented research on identity development in the nursing profession, Fischer [33] asks, among other things, whether student nurses think about what they can change, to what extent student nurses support each other, or whether they are interested in how their actions contribute to the overall events in a hospital. The main category 4.1 “Topics and key questions” can provide a useful qualitative supplement to these questions. 

In her quantitatively oriented study, Roddewig [23] focuses on the value of collegial advice on the emotional well-being of student nurses. Among other things, she looks at sensitivities and coping strategies. Especially the categories found here within the main category “2.3 Well-being” could provide a useful complement to the importance of sensitivities and the effectiveness of collegial counselling as a coping strategy. Furthermore, Roddewig examines concrete indicators of the success of collegial consultation (e.g., spacing of consultations, self-led vs. facilitated groups, perceived effectiveness, and usefulness of collegial consultation). The category system presented here takes similar points in the superordinate categories 1 “Organizational conditions,” 2 “Didactic conditions,” and 3 “Recommendations for the implementation of collegial advice” and could confirm or challenge the results already available. In addition, Roddewig presents a training program that includes several methods of advising. These overlap only in one method with the methods listed in category “2.2 Consulting Methods”. Thus, the system represents an extension of the method repertoire, which can lead to a contrasting discussion. 

Other concrete counselling methods are suggested, by Grässlin and Fallner [34], Tietze [35], Lippmann [18] or Hendriksen and Huizing [36], for example. However, these authors work more on a descriptive level and not concretely in the field of vocational training. With the category system, these methods, applied in the system of nursing education, could be considered, and evaluated qualitatively for the first time. 

With the category system it is possible to make subjectively perceived competence development and the influences on it visible. Furthermore, framework conditions of collegial consultation and concrete methods can be evaluated. The still pending analysis of the data will show whether it can live up to this claim.

## Figures and Tables

**Figure 1 healthcare-10-02517-f001:**
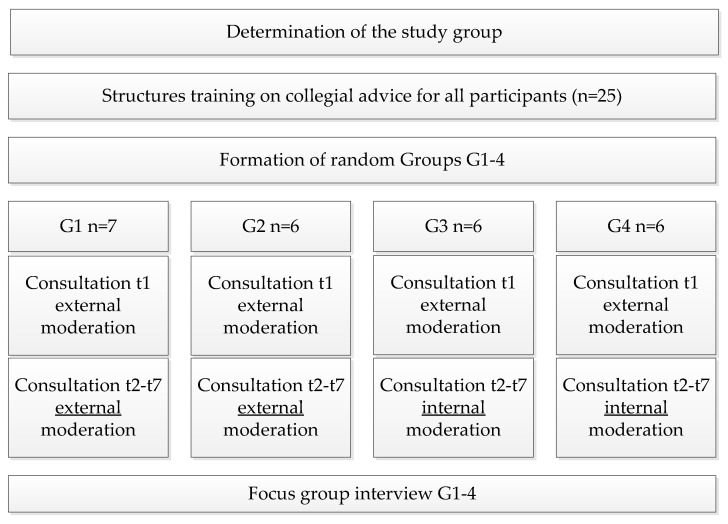
Procedure of the intervention.

**Figure 2 healthcare-10-02517-f002:**
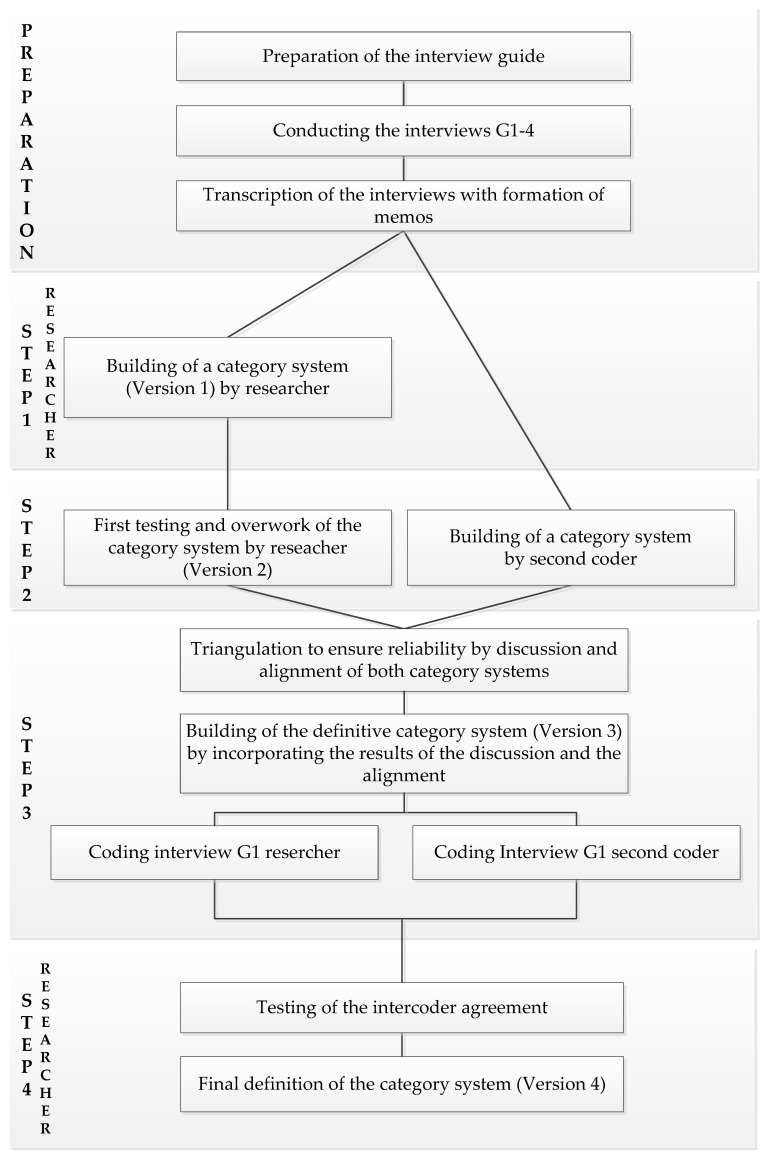
Process of creating the category system.

**Figure 3 healthcare-10-02517-f003:**
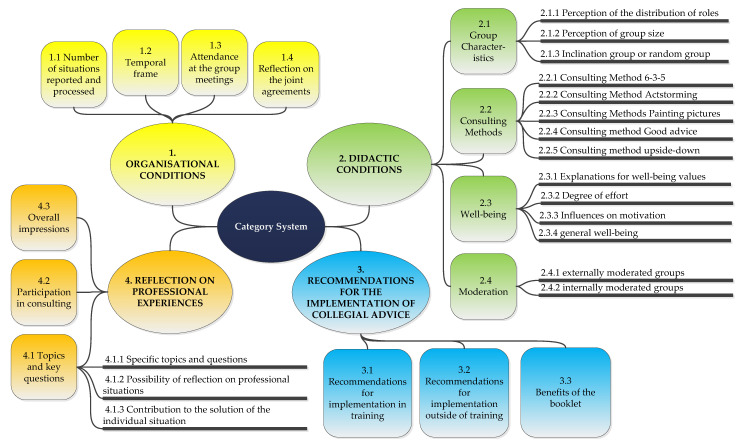
Overview of the category system.

**Table 1 healthcare-10-02517-t001:** Category system with selected anchor examples.

Name of the Category	Type of Category	Anchor Examples
1. ORGANISATIONAL CONDITIONS	Superordinate category	Always empty
1.1 Number of situations reported and processed	main category	“We talked so much about our problems that we first briefly told all the situations and then at the end we managed only one consultation.” [Participant 21]
1.2 Temporal frame	main category	“The time was reasonable, and we usually managed two situations well. If one of them took a little longer, that was also fine.” [Participant 21]
1.3 Attendance at the group meetings	main category	“It is noticeable that one person was not there so often, only four times.” [Participant 17]
1.4 Reflection on the joint agreements	main category	“I place a lot of value on respectful interaction. No matter what someone says, you shouldn’t laugh about it or make fun of it. That what is said is respected and accepted, that is very important to me.” [Participant 19]
2. DIDACTIC CONDITIONS	superordinate category	Always empty
2.1 Group characteristics	main category	Empty
2.1.1 Perception of the distribution of roles	subcategory	“I felt comfortable in both roles because I thought: I’ll use this for myself. And when I have a problem and others give me something, I like to take that with me. I also like to give advice.” [Participant 21]
2.1.2 Perception of group size	subcategory	“I found it easy to talk about things because we were such a small group. In larger groups, some things are more difficult to discuss. I wouldn’t dare tell anything there.” [Participant 17]
2.1.3 Inclination group or random group	subcategory	“I think it’s an advantage when a group is drawn. You hear opinions from people you wouldn’t otherwise ask.” [Participant 21]
2.2 Consulting methods	main category	empty
2.2.1 Consulting method 6-3-5	subcategory	“I think the ‘6-3-5 method’ is the best way to get your advice across. The way you really mean it.” [Participant 23]
2.2.2 Consulting method Actstorming	subcategory	“’Actstorming’ did not go over well with our group.” [Participant 21]
2.2.3 Consulting methods Painting pictures	subcategory	“The other methods were easier to get advice across.” [Participant 23]
2.2.4 Consulting method Good advice	subcategory	“I think the ‘good advice’ method is the best way to get your advice across. The way you really mean it.” [Participant 23]
2.2.5 Consulting method Upside-down	subcategory	“I think ‘upside-down’ is a method that helps when you get stuck. If you have no ideas, you can definitely say what you should not do. This leads to possibilities of what you could do. Whether this is always useful, I do not know.” [Participant 21]
2.3 Well-being	main category	empty
2.3.1 Explanations for well-being values	subcategory	“After the vacation or after an interval of free time, I was fine. And then I noticed that my well-being questionnaires turned out much better than before.” [Participant 22]
2.3.2 Degree of effort	subcategory	“I found it was hard to remember an example and present it as a case or problem when it had been in the past for a while.” [Participant 22]
2.3.3 Influences on motivation	subcategory	“When we did the consultation in the morning, it was better. In the afternoon, I was pretty knocked out. I didn’t feel like discussing things anymore.” [Participant 17]
2.3.4 General well-being	subcategory	“Overall, I think the values are very poor.” [Participant 17]
2.4 Moderation	main category	Empty
2.4.1 Externally moderated groups	subcategory	“I find externally moderated is better.” [Participant 22]
2.4.2 Internally moderated groups	subcategory	Empty
3. RECOMMENDATIONS FOR THE IMPLEMENTATION OF COLLEGIAL ADVICE	superordinate category	Always empty
3.1 Recommendations for implementation in training	main category	“The consultations were helpful for me. I would have liked the collegial advice sooner, in the first year of teaching, before it all starts.” [Participant 18]
3.2 Recommendations for implementation outside of training	main category	“I would like to see collegial consultation continued after the training.” [Participant 22]
3.3 Benefits of the booklet	main category	“I found the booklet very helpful, especially in the beginning. I didn’t remember many of the methods. I got the methods mixed up. In the end, I didn’t need the booklet anymore.” [Participant 19]
4. REFLECTION ON PROFESSIONAL EXPERIENCES	superordinate category	Always empty
4.1 Topics and key questions	main category	“Sometimes I have problems that I think are petty and that I don’t need to discuss.” [Participant 22]
4.1.1 Specific topics and questions	subcategory	„Often, these are topics that have to do with the team. Whether it’s your Mentor, colleagues, or ward manager.” [Participant 22]
4.1.2 Possibility of reflection on professional situations	subcategory	“There were examples where you were no longer in the situation. But I still found the results of the consultation helpful. For example, if you get into a similar situation.” [Participant 21]
4.1.3 Contribution to the solution of the individual situation	subcategory	“We solved problems through different methods, and I found these methods good and varied.” [Participant 20]
4.2 Participation in consulting	main category	“I think it’s a shame that not everyone in the group contributed so much. I would have liked to hear more from some.” [Participant 21]
4.3 Overall impressions	main category	“I liked the collegial advice very much.” [Participant 17]

**Table 2 healthcare-10-02517-t002:** Complete category description of the Category 4.1.2.

Number and path 4.1.2 Reflection on professional experiences\Topics and key questions\Possibility of reflection on professional situations
Name of the category Possibility of reflection on professional situations
Description of the category This category summarizes content that indicates whether collegial advice is generally suitable for reflecting on professional experiences. Concrete examples are coded as well as general statements about the possibility of reflecting on professional experiences. Content on the degree of assistance provided by collegial consultation is also included in the coding. This is explicitly about the perception of whether collegial advice seems fundamentally suitable or unsuitable. The objective is to find out whether the participants accept or reject collegial advice as a solution approach for dealing with professional situations.
Application of the category This category is used when references are made to the possibility of reflecting on the professional situation.
Example of application „There were examples where you were no longer in the situation. But I still found the results of the consultation helpful. For example, if you get into a similar situation.” [Participant 21]
Differentiation from other categories (optional) empty

## Data Availability

The data presented in this study are available on request from the corresponding author.

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
