# Peer review of "Designing Categories for a Mixed-Method Research on Competence Development and Professional Identity through Collegial Advice in Nursing Education in Germany"

_healthcare, 2022, doi:10.3390/healthcare10122517_

Round 1

Reviewer 1 Report

Thank you for the opportunity to review the manuscript. Overall, a special topic for concerning the development of nursing education and further exploration of this topic is certainly interest, especially to examine how collegial consultation affects the development of competence and professional identity of student nurses in Germany.

A few questions / comments and suggestions:

In Line 50, describe the “One’s own work”, relevance to the study is not clear.

In Line 80, what is the meaning for “reciprocity of all roles”, relevance to the study is not clear.

In Line 101, refine another word of “Article” for this meaning.

In Line 173, how to generate interview questions, relevance to the study is not clear.

In Figure 2 Process of creating the category system, elaborate more detail in the box stated as “building of the definitive category system (Version 3)”.

In Line 256-262, the reasons for discrepancies in coding described is not very clear to be identified how the authors moved from point form to the themes presented.

In Line 277, Table 1, please edit for the table content and format more clearly.

In Line 294, described “Very good results were obtained here, relevance to the study is not clear.

In Line 300, described “Some categories”, relevance to the study is not clear.

In Line 306, “The validity of the instrument seems to be high here”, relevance to the study is not clear.

In Line 307, the quantitative data result not shown in this manuscript, relevance to the study is not clear.

In Line 316-318, relevance to the study is not clear.

In Line 324-325, how to connect this development, relevance to the study is not clear.

In Line 346-348, relevance to the study is not clear.

Overall of conclusion of this study is not clear and suggest setting another session mainly to demonstrate the conclusion of this study.

Thank you for the opportunity to review the manuscript. Overall, a special topic for concerning the development of nursing education and further exploration of this topic is certainly interest, especially to examine how collegial consultation affects the development of competence and professional identity of student nurses in Germany.

A few questions / comments and suggestions:

In Line 50, describe the “One’s own work”, relevance to the study is not clear.

In Line 80, what is the meaning for “reciprocity of all roles”, relevance to the study is not clear.

In Line 101, refine another word of “Article” for this meaning.

In Line 173, how to generate interview questions, relevance to the study is not clear.

In Figure 2 Process of creating the category system, elaborate more detail in the box stated as “building of the definitive category system (Version 3)”.

In Line 256-262, the reasons for discrepancies in coding described is not very clear to be identified how the authors moved from point form to the themes presented.

In Line 277, Table 1, please edit for the table content and format more clearly.

In Line 294, described “Very good results were obtained here, relevance to the study is not clear.

In Line 300, described “Some categories”, relevance to the study is not clear.

In Line 306, “The validity of the instrument seems to be high here”, relevance to the study is not clear.

In Line 307, the quantitative data result not shown in this manuscript, relevance to the study is not clear.

In Line 316-318, relevance to the study is not clear.

In Line 324-325, how to connect this development, relevance to the study is not clear.

In Line 346-348, relevance to the study is not clear.

Overall of conclusion of this study is not clear and suggest setting another session mainly to demonstrate the conclusion of this study.

Author Response

Dear Reviewer, 

Kind regards, 

Stefan Wellensiek

Reviewer 2 Report

Very good article. The research method is described very carefully and makes a clear case for the style of education in Germany. I do think the article could include a deeper discussion of research results or some comment comparing the style of education in Germany to that of a different country, but this is not a major concern. Support publication of the article with minor edit/spell check.

Author Response

(The authors gave the same response as above.)

Reviewer 3 Report

Dear Authors

it is very crucial.

Please can you describe if there are gender differences or differences related to social economic status?

Author Response

(The authors gave the same response as above.)

Reviewer 4 Report

Dear authors.

Thank you for the opportunity to review this paper and congratulations on your work.

The manuscript aims to determine how collegial consultation affects the development of competence and professional identity of student nurses, designing categories for mixed-method research on competence development and professional identity through collegial advice in nursing education, in Germany.

It was a very readable article, with a very well-explained process and well accompanied by figures and references. The conclusions are supported by the explanations, as are the criteria of objectivity, reliability, and validity.

To take a step beyond the initial approach, which I understand is not the aim of the article but could be an approach for subsequent ones, perhaps to have conducted an initial form focused on the knowledge of the students with collegial consultation and to repeat it in Phase three.

Finally, has the collegial consultation been implemented or do we have real data on its usefulness? I think it would be very interesting to export it to other countries.

Regards.

Author Response

(The authors gave the same response as above.)

Reviewer 5 Report

Many thanks for the opportunity to read this interesting manuscript about development of nursing identity and competence development. Considering that in the Germany academization of the nursing profession is very much work in progress, this manuscript provides an insight about how peer advice process may contribute to the process of developing skills and professional competencies among nurse students. Overall the manuscript was well written, but especially question setting should be clarified as throughout the manuscript there was hinting about what was to come – distracting about the current results included in this manuscript. In addition, the discussion would have benefited from further reflection about the results in context of nurse education and development of professional skills and identity.

Abstract

In the methods par it is stated that “qualitative part” – What other “parts” were involved? If any.

Abstract seems to be missing a clear discussion/conclusions sections. This should be added.

Introduction

Line 31: This is unclear – currently this sentence reads as nurses were previously not expected to acquire skills. Do authors refer to advanced skills? Please give an example.

Line 36: Repeating the word professional. Do the authors refer here complex nursing or healthcare situations?

Line 46: “no competence development would be conceivable” – why isn’t competence development possible without professional identity? Please could you also add a reference.

Line 67: Is something (a word) missing from the quotation?

Line 74: Specifying authors without any information e.g. names of the models feels a bit pointless.

Line 81: “Collegial advice, also known as intervision” – repeat from line 76.

Line 99: Are collegial advice and collegial counselling used as synonyms or two different methods? This is unclear.

Question setting should be clarified. It appears that two different research questions are examined within the MS. However, authors -somewhat confusingly – state that “Before answering this, there is the task of evaluating qualitative data that is collected.”. This leaves it unclear what it the focus in this MS.

Methods

Participants

Authors have not commented about the importance/non-importance of how well the colleagues participating in the process should be acquainted with each other’s. Or does this matter. This appears an important aspect, as here it is emphasized that the students were well acquainted with each other’s.  

Data Collection

While considerable amount of information is given about the collected data, authors should make it clearer what data was collected from records during the group meetings (who recorded this?) and what was collected from focus group interviews. Considering the question setting, it would be also good to guide the reader what data collection was orientated to answer which question.

Data processing

Please could the authors extent this section by describing e.g. how memos and codes were formed i.e. what were easy to follow rules. It is also rather cryptically stated that “To avoid negative effects on the qualitative research,” – please could the authors explain what negative effects are referred to in here.

Line 172: there seems to be unnecessary repeat of “category formation.

Line 205: Sentences shouldn’t start with a numeric expression – should be written e.g. Twenty-three.

Discussion

Apart from reflecting of the objectivity, reliability and validity of the constructs, the discussion would have benefited in additional reflection in how this relates to development of professional identity among student nurses.

Author Response

(The authors gave the same response as above.)

Round 2

Reviewer 3 Report

FOR ME IT IS REJECT
